# Andean Sacha Inchi (*Plukenetia Volubilis* L.) Leaf-Mediated Synthesis of Cu_2_O Nanoparticles: A Low-Cost Approach

**DOI:** 10.3390/bioengineering7020054

**Published:** 2020-06-06

**Authors:** Brajesh Kumar, Kumari Smita, Alexis Debut, Luis Cumbal

**Affiliations:** 1Post Graduate Department of Chemistry, TATA College, Kolhan University, Chaibasa, Jharkhand 833202, India; 2Centro de Nanociencia y Nanotecnologia, Universidad de las Fuerzas Armadas ESPE, Av. Gral. Rumiñahui s/n, Sangolqui P.O. BOX 171-5-231B, Ecuador; kumarismi@gmail.com (K.S.); apdebut@espe.edu.ec (A.D.); lhcumbal@espe.edu.ec (L.C.)

**Keywords:** *Plukenetia volubilis*, green synthesis, Cu_2_O nanoparticles, XRD, catalysis

## Abstract

In this work, Andean sacha inchi (*Plukenetia volubilis* L.) leaves were used to prepare monodispersed cuprous oxide (Cu_2_O) nanoparticles under heating. Visual color changes and UV-visible spectroscopy of colloidal nanoparticles showed λ_max_ at 255 nm, revealing the formation of copper oxide nanoparticles. Transmission electron microscopy and dynamic light scattering analysis indicated that the prepared nanoparticles were spherical with an average size of 6–10 nm. The semi-crystalline nature and Cu_2_O phase of as-prepared nanoparticles were examined by X-ray diffraction. Fourier-transform infrared spectroscopy confirmed the presence of polyphenols, alkaloids and sugar in the sacha inchi leaf, allowing the formation of Cu_2_O nanoparticles from Cu^2+^. Additionally, as-synthesized Cu_2_O nanoparticles exhibited good photocatalytic degradation activity against methylene blue (>78%, 150 min) with rate constant 0.0219106 min^−1^. The results suggested that the adopted method is low-cost, simple, ecofriendly and highly selective for the synthesis of small Cu_2_O nanoparticles and may be used as a nanocatalyst in the future in the efficient treatment of organic pollutants in water.

## 1. Introduction

During the last two decades, nanoparticles/nanomaterials (of size less than 100 nm) have been extensively studied due to their exclusive properties such as high surface-to-volume ratio, flexibility, quantum size, high yield strength, rigidity, ductility and macro-quantum tunneling effect, and are currently used in various areas of chemistry, physics, medicine and engineering [1,2]. Copper is one of the most important elements used worldwide for various purposes. In particular, copper oxide nanoparticles have received great attention because of their low cost, high yield, mild reaction conditions and fantastic applications in batteries [3], catalysis [4], optical devices [5], printed electronics [6], anticancer therapeutics, sensing, antioxidants [7], antimicrobial activities [8], fuel cells [9], bioimaging [10], dye removal [11], gas sensors [12], etc.

Copper oxide nanoparticles have been synthesized by a variety of chemical and physical methods including hydrothermal synthesis [13], the wet chemical method [14], solution phase synthesis [15], sonochemical synthesis [16], the microwave method [17], the laser ablation method [18] and ball milling [19], in which a large amount of solvents is required for obtaining pure and well-defined nanoparticles. They create various problems for the ecosystem and the environment [7,20]. Plant-based metal nanoparticle preparation methods are low-cost, simple and ecofriendly and hold great promise in biotechnology applications due to the presence of effective reducing and capping agents such as carbohydrates, polyphenols, terpenoids, sugars, amino acids, flavanoids, saponins, etc. [20]. Therefore, they are favored over chemical and physical synthesis methods [21]. Previously reported syntheses of cuprous oxide/cupric oxide (Cu_2_O/CuO) nanoparticles have used plant materials to support large-scale synthesis, including extracts of *Azadirachta indica* leaf [22], *Aegle Marmelo* leaf [23], *Aloe barbadensis Miller* leaf [24], *Acalypha indica* leaf [25] *Punica granatum* peel [26], *Terminalia arjuna* bark [27], *Stachys lavandulifolia* flower [28], *Terminalia bellirica* [29] and *Rubus glaucus* fruit [30], etc.

Sacha inchi (SI) (*Plukenetia volubilis* L.) is a promising crop plant originally from the Amazon basin of Latin America. Its star-shaped green fruits produce nut-like seeds with a bitter taste and have been consumed by humans since Incan times due to their high content of fatty acids (ω-3 and ω-6, 35–60%), protein (27–33%), carbohydrates and antioxidants [31]. Its dried leaves are marketed as a tea and contain 84.2–93.4% sugar [32]. Phytochemicals such as saponins, terpenoids, polyphenolic compounds (flavonoids) and other components are also found in SI leaves and are responsible for their antioxidant and antiproliferative activities [33].

Thus, the utilization of these lost crops in nanotechnology and green chemistry has been continued by our group: SI leaves for the synthesis of silver nanoparticles [34], SI oil for the synthesis of silver [35] and gold nanoparticles [36] and SI shell biomass for the synthesis of silver nanoparticles [37] and removal of Cu^2+^/Pb^2+^ [38]. Our research on the SI plant may support an additional source of revenue for farmers on the western and northern edges of South America, such as Colombia, Ecuador, Venezuela, Peru, Bolivia, Brazil and Suriname, and in the Lesser Antilles [31].

To our knowledge, there have been no reports of the synthesis of Cu_2_O nanoparticles using the SI leaf. Herein, green Cu_2_O nanoparticles were synthesized by reducing Cu^2+^ ions with aqueous extract of SI leaf. This nanoparticle is used in the degradation of methylene blue (MB), one of the most common organic pollutants in wastewater from the dye industry (Figure 1). It is environmentally undesirable at a trace level and excessive usage of MB causes serious diseases in human beings [35,36]. Hence, the development of an ecofriendly method for the removal of MB from wastewater is necessary. In this work, full spectroscopic and microscopic characterizations were performed to confirm the formation of Cu_2_O nanoparticles. Photocatalytic evaluation experiments indicated that Cu_2_O nanoparticles possess fast catalytic activity in the degradation of MB.

## 2. Materials and Methods

### 2.1. Materials

Copper nitrate (Cu(NO_3_)_2_·3H_2_O, 99.0%) and methylene blue (MB, 99.5%) were purchased from Spectrum, USA. Sacha inchi (SI) leaves were collected with the help of Mr. Abraham Rodolfo Sanchez Piňuela from a farm in Quito, Ecuador. Milli-Q water was used in all experiments. All chemicals listed above were of analytical grade and used without any purification.

### 2.2. Preparation of Cu_2_O Nanoparticles

Thoroughly washed SI leaves were shade-dried for one week. The extraction of phytochemicals from the SI leaves was performed by an earlier extraction method [34]. Nearly 800 mg of the dried SI leaf was crushed into small pieces and transferred to a 100 mL flask containing 50 mL of Milli-Q water, and then it was mixed well and heated (64–68 °C) for 30 min. The obtained red extract of SI leaf was filtered through Whatman No. 1 paper and stored at 4 °C for further use. For the synthesis of Cu_2_O nanoparticles, 5 mL of SI leaf extract was added slowly to 20 mL of Cu(NO_3_)_2_ solution (10 mM), followed by heating for 5 h at 85–90 °C with continuous stirring. The formation of the Cu_2_O nanoparticles was indicated by a change of reaction mixture color from bluish-red to a greenish color.

### 2.3. Characterization of Cu_2_O Nanoparticles

The samples containing nanoparticles were confirmed by a UV-visible spectrophotometer, GENESYS^TM^ 8 from Thermo Spectronic, England. The particle size distribution of the sample was analyzed using the HORIBA, DLS Version LB-550 program, Japan. The morphology and selected area electron diffraction (SAED) pattern of the nanoparticles were captured on a transmission electron microscope (TEM), FEI Tecnai, G2 Spirit Twin, Holland. X-ray diffraction (XRD) studies on thin films of the nanoparticle were carried out using a PANalytical brand θ-2θ configuration (generator-detector) X-ray tube, copper λ = 1.54059 Å, and an EMPYREAN diffractometer. The Fourier-transform infrared (FTIR) spectra were collected on a Spectrum Two IR spectrometer from Perkin Elmer, USA. The samples for FTIR and XRD analysis were prepared by carefully depositing a thin film of Cu_2_O nanoparticles on a glass slide by injecting and heating 1800 μL (600 μL × 3 times) of Cu_2_O solution drop by drop at 60–65 °C for 20–30 min allowing the solvent to evaporate. After that, the thin film of the samples was scratched and the FTIR-ATR analysis was performed.

### 2.4. Photocatalytic Effects

The photocatalytic activity of Cu_2_O nanoparticles during the degradation of MB was determined by carrying out the reaction in direct sunlight (1040–1165 cd/m^2^) at 30–35 °C (atmospheric temperature). Typically, 5 mL MB (10 mg/L) was mixed with 1 mL H_2_O in a glass tube; in a second glass tube, 5 mL MB (10 mg/L), aqueous solution containing Cu_2_O nanoparticles (500 μL) and H_2_O (500 μL) was mixed in the dark for 20 min to reach an adsorption-desorption equilibrium. Then, both sets were exposed to direct sunlight and the progress of the degradation reaction was monitored at different time intervals by UV-vis spectroscopy at a wavelength of 664 nm. To evaluate the interference of SI leaf extract on the reduction of MB, a separate reaction was performed in which 5 mL of MB (10 mg/L) was mixed with 500 μL of SI extract and 500 μL of H_2_O. After that, the reaction mixture was heated at 40 °C for 120 min in the dark. The photocatalytic degradation percentage of MB was calculated using Equation (1) and the respective first-order rate constants (k) according to Equation (2).
η = (A_0_ − A_t_)/A_0_ × 100%(1)
kt = ln (A_0_/A_t_)(2)
where η is the rate of degradation of MB in terms of percentage, A_0_ is the initial absorbance of the dye solution and At is the absorbance of the MB at time t, respectively. C_0_/C_t_ is measured from the relative intensity of absorbance (A_0_/A_t_). The linear relationship of ln (A_0_/A_t_) versus time indicates that the photodegradation of MB follows first-order kinetics [35,36].

## 3. Results and Discussion

### 3.1. UV-Vis Spectroscopy Analysis

Figure 2 shows the UV-visible absorption spectra of as-prepared Cu_2_O nanoparticles in the presence of SI leaf extract in an aqueous medium. The colorimetric reduction reaction of Cu^2+^ ions with the SI leaf extract (red colour) indicated the formation of a greenish-colored solution (a and b in Figure 2), attributed to the surface plasmon resonance (SPR) of the Cu_2_O nanoparticles [30,39]. The absorption peaks appearing at 260 and 340 nm (red line) correspond to the polyphenolic compounds present in the SI leaf extract [34], while as-prepared Cu_2_O nanoparticles exhibited a broad absorption peak at between 240 to 380 nm having two λ_max_ at 255 and 360 nm (green line), respectively [39,40]. The change observed in the spectrum after the reduction of Cu^2+^ ions to Cu_2_O nanoparticles corresponds to the formation of Cu-phytochemicals, complex or spherical Cu_2_O nanoparticles with size 2–5 nm and quite stable in the shape-position of absorption after one month, indirectly indicating the stability of the nanoparticles [39,40].

### 3.2. TEM and SAED Analysis

The exact shape and size of the nanoparticles was determined using a TEM image. The high- (Figure 3a) and low- (Figure 3d) magnification TEM images showed a good dispersion with the spherical morphology of Cu_2_O nanoparticles inside the SI leaf extract. The majority of the nanoparticles observed from the TEM micrograph are small, with a size of 6–10 nm, indicating the availability of high surface catalytic activity for Cu_2_O nanoparticles; a small few of the Cu_2_O nanoparticles are of a bigger size, around 20–45 nm. This may be due to the presence of excessive SI phytochemicals on the surface of the Cu_2_O nanoparticles, which can cause the aggregation and enlargement of nanoparticles during the synthesis of Cu_2_O nanoparticles. The size distribution pattern of the nanoparticles observed in the TEM image (Figure 3a) was analyzed manually using ImageJ software. It showed the average size of the Cu_2_O nanoparticles to be 6.67 ± 3.96 nm (Figure 3b). No aggregation of nanoparticles suggested the presence of hydrophobic coating around the Cu_2_O nanoparticles. The SAED pattern of the Cu_2_O nanoparticles (Figure 3c) shows partial concentric rings which signify that the particles are spherical shape, semicrystalline nature and indexed as (111) and (200) lattice planes for the face-centred cubic (FCC) structure of Cu_2_O nanoparticles [16].

### 3.3. DLS Analysis

The dynamic light scattering (DLS) technique gives valuable information about the hydrodynamic size distribution of the Cu_2_O nanoparticles (Figure 4). The average width of the Cu_2_O nanoparticles was found to be 8.2 ± 2.1 nm with polydispersity index (PDI) = 0.0656. The observed PDI < 0.1 clearly indicates that the synthesized Cu_2_O nanoparticles were monodispersed in nature [29]. However, the average size of as-synthesized Cu_2_O nanoparticles determined by the DLS method was slightly higher than in the TEM analysis. This is due to either the presence of organic coating around nanoparticles or the screening of small particles by bigger ones [36,41].

### 3.4. XRD Analysis

In order to clarify the nature and crystallinity of the Cu_2_O nanoparticles, XRD analysis was performed as shown in Figure 5. The presence of intense peaks at 2θ = 36.46 and 42.45° represents the (111) and (200) planes. The XRD reflections of Cu_2_O match that of Inorganic Crystal Structure Database ICDD (Inorganic Crystal Structure Database) no. 98-018-0846 corresponding to the semi-crystalline cubic FCC structure, which match the common peaks in the earlier report [42]. It can be also observed that the undesired peak at 2θ = 32° is due to a typical impurity caused by a metallo-organic structure [23,24]. It is likely that these peaks indicate that the phytochemicals in SI leaf extract attached to the Cu_2_O nanoparticles and were also involved with surface-capped nanoparticles. However, the observed size of the nanoparticles in the XRD using the Scherrer formula (~46 nm) is bigger than in the TEM-DLS results [40]; this may be due to the aggregation of particles during drying for the preparation of XRD samples. This increment could be attributed to the light scattering effect because of the aggregation of the metal [43].

### 3.5. FTIR Analysis

FTIR spectra of the particles were recorded to detect the chemical interaction between SI leaf extract and Cu^2+^ that occurs during the synthesis of Cu_2_O nanoparticles (Figure 6). They reveal a weak absorption band at 698 cm^−1^ may correspond to the Cu-O bond vibrational frequencies, which are slightly higher than reported Cu–O (645 cm^−1^) bond vibration due to depending on the degree of hydrogen bonding. Furthermore, the additional Cu–O–H bonds lead to bending absorptions in the region of 896 cm^−1^ [44]. This can be attributed to the presence of the Cu–O bond. The strong band at 1028 cm^−1^ can be attributed to the C–O–C/secondary C–OH bonds in the polysaccharide/protein structure of the SI leaf [34]. The prominent bands at around 1241, 1314, 1374 and 1601 cm^−1^ can be attributed to the vibrational mode for C–H, C–N, C=C and C=O (amide 1) [30]. The existence of the peaks at 2847 and 2921 cm^−1^ is due to the symmetric and asymmetric C–H stretching vibrations of flavonoids/phenolic compounds, respectively. The broad absorption peak at 3331 cm^−1^ shows the existence of O–H/N–H stretching groups of macromolecular association [32,34]. The FTIR results indicate that the extract of SI leaf acted as a reducing and capping agent in the synthesis of the Cu_2_O nanoparticles.

### 3.6. Photocatalytic Activity

Figure 7a shows the photocatalytic activity of Cu_2_O nanoparticles in the degradation of MB. The peaks at around 300 nm correspond to the absorption of the benzene ring and the peaks between 600–700 nm represent the absorption of heteropoly aromatic linkage of MB [45]. It can be seen that the degradation of MB increases with increasing solar irradiation time by observing the decrease in λ_max_ at 664 nm [35]. The photodegradation percentages (η) of MB for 30, 75 and 150 min are 25.45%, 59.88% and 78.90% in the presence of Cu_2_O nanoparticles when compared with that of the respective controls. No change of absorbance at λ_max_ = 664 nm was observed when a blank test was performed with MB and SI leaf extract. This confirms that the SI leaf extract does not interfere with the reduction of MB. In Figure 7b, the plots of ln (A_0_/A_t_) versus time yield good linear correlations and are well fitted with pseudo-first-order kinetics. The result indicates that the observed rate constant (k) and correlation coefficient (R^2^) for the MB degradation were 21.9106 × 10^−3^ min^−1^ and 0.9999. Hence, the present study highlights the promising potential of SI leaf-synthesized Cu_2_O nanoparticles for MB degradation in wastewater.

## 4. Conclusions

In conclusion, the use of SI leaf as a bioreducant for the synthesis of Cu_2_O nanoparticles is a low-cost, simple and ecofriendly approach. The results obtained from DLS, TEM, XRD and FTIR suggest that the Cu_2_O nanoparticles are highly dispersed, spherical, 6–10 nm in size, semicrystalline and surface-capped with SI leaf extract. The applied approach provided a Cu_2_O nanocatalyst, which presented very good catalytic performance for the degradation of MB under sunlight.

## Figures and Tables

**Figure 1 bioengineering-07-00054-f001:**
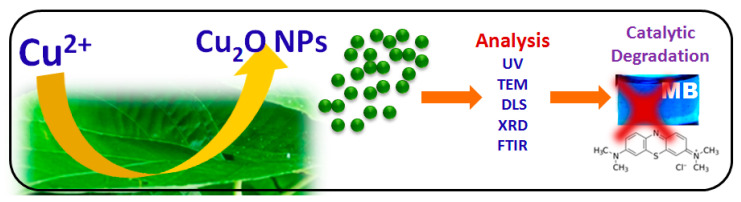
Schematic presentation of the synthesis and application of Cu_2_O nanoparticles.

**Figure 2 bioengineering-07-00054-f002:**
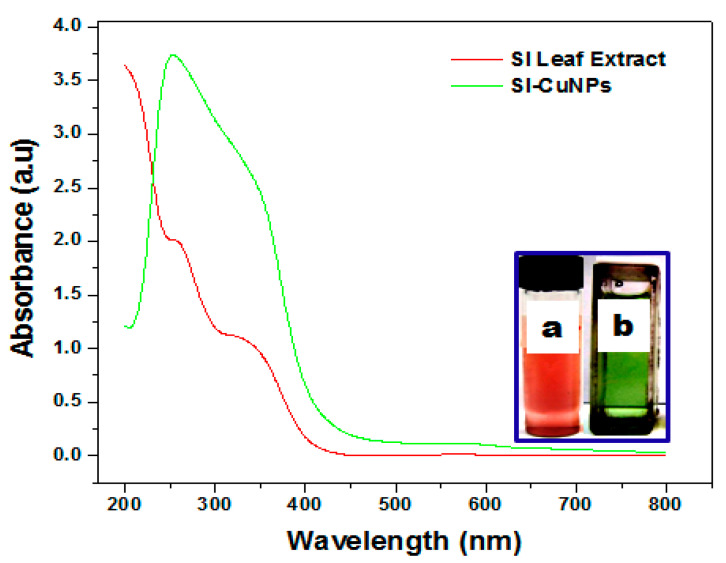
UV-vis spectra of SI leaf extract (red line) and Cu_2_O nanoparticles (green line). Visual picture of solution containing (a) SI leaf extract and (b) Cu_2_O nanoparticles.

**Figure 3 bioengineering-07-00054-f003:**
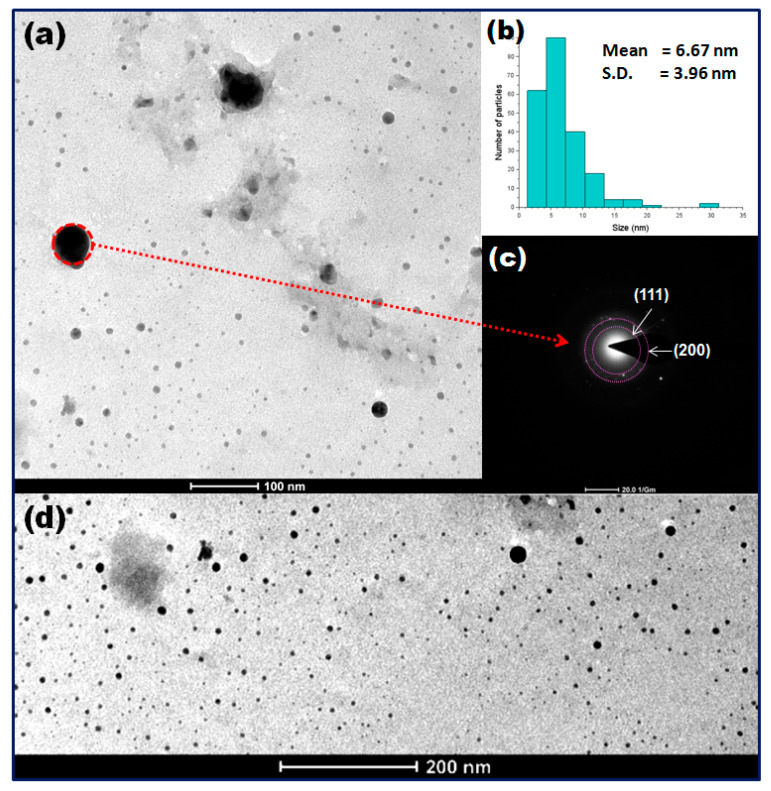
(**a**,**d**) TEM (transmission electron microscope) images of Cu_2_O nanoparticles synthesized in solution, (**b**) size distribution pattern and (**c**) SAED pattern.

**Figure 4 bioengineering-07-00054-f004:**
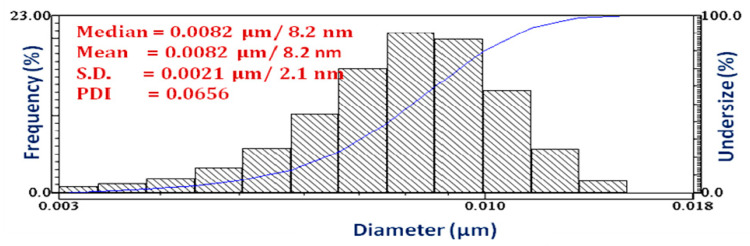
DLS size distribution of the Cu_2_O nanoparticles. Abbreviations: DLS, dynamic light scattering.

**Figure 5 bioengineering-07-00054-f005:**
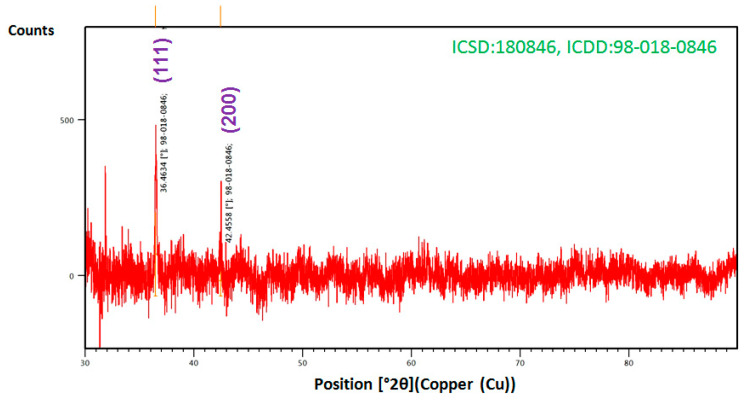
XRD pattern of the Cu_2_O nanoparticles. Abbreviations: XRD, X-ray diffraction.

**Figure 6 bioengineering-07-00054-f006:**
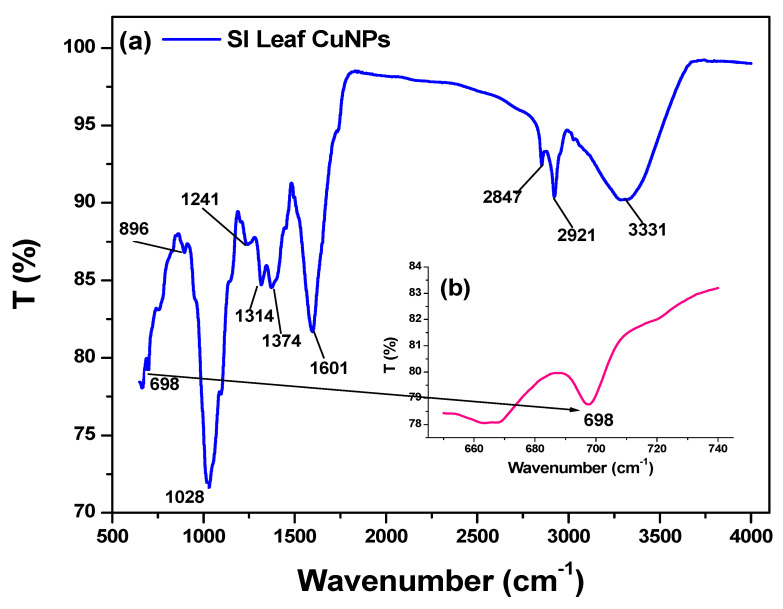
(**a**) FTIR spectrum of as-synthesized Cu_2_O nanoparticles using SI leaf extract and (**b**) specific Cu–O at 698 cm^−1^.

**Figure 7 bioengineering-07-00054-f007:**
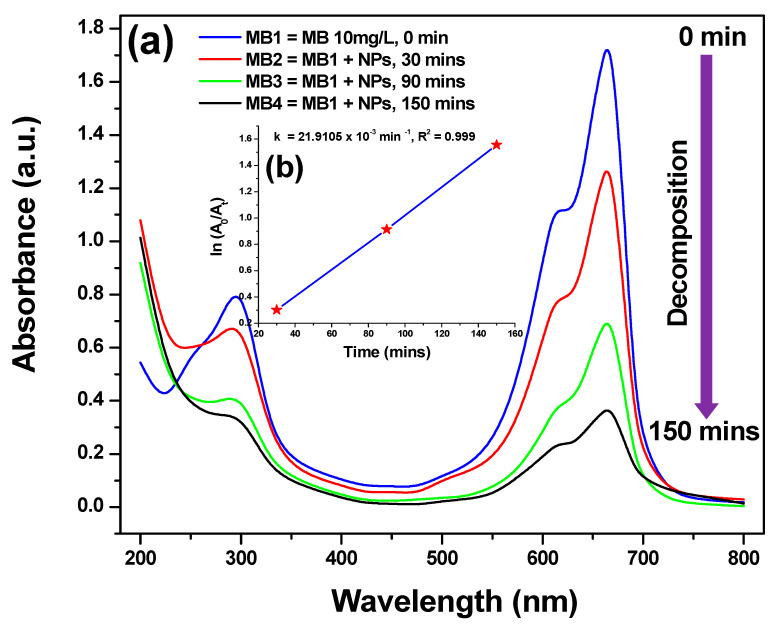
(**a**) Photocatalytic degradation pattern and (**b**) ln (A_0_/A_t_) vs. time kinetic plot of MB using Cu_2_O nanoparticles as photocatalyst.

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
