# Peer review of "Andean Sacha Inchi (Plukenetia Volubilis L.) Leaf-Mediated Synthesis of Cu2O Nanoparticles: A Low-Cost Approach"

_bioengineering, 2020, doi:10.3390/bioengineering7020054_

Round 1

Reviewer 1 Report

Kumar and coworkers have described a method for green synthesis of copper oxide nanoparticles using sacha inchi leaves extract. While the results are incremental to the previous studies in the field, I do not find them novel enough for publishing in bioengineering journal. Moreover, authors need to perform additional studies to further validate the potential applications of the synthesized nanoparticles.  I have listed some of my concerns below-:

  1. The TEM image should have more number of nanoparticles (ideally more than 100) to demonstrate homogenous nanoparticles.
  2. The authors should compare the stability of leaf extracts produced copper-oxide nanoparticles with other conventionally synthesized Cu2O NPs.

Authors need to demonstrate the potential applications of the synthesized Cu2O NPs and also stress on their advantages over other green synthesis methods. Overall, the manuscript is not suitable for publication in bioengineering.     

Reviewer 2 Report

This is an interesting work, describing how plant extract from Sacha inchi can be used as a reducing and capping agent for the synthesis of Cu2O nanoparticles. The authors further uses the as synthesized Cu2O and sunlight for degrading methylene blue. I believe this work should accepted for publication after some minor corrections. 

Introduction section:

Since the authors describe the Sacha inchi for 'additional source of revenue for farmers worldwide' but originate from the Amazon basin, it will be nice to actually indicate some places in the world that it is reportedly found aside the Amazon basin, and in particular, could they add a reference?

The SI must defined in the L54 the first time it is used in the main text, and before it can be used in L59 and thereafter. 

Can the authors kindly state the importance of the chosen catalytic application, e.i. why is degradation of MB important?

Experimental section:

There are generally some grammatical errors that the authors will like to fix. It will read clearer if the author can be consistent in the tenses used in describing the procedures. eg L82-83, L86-87. 

In the catalytic testing, what state or form was the Cu2O np before adding 500 uL water? Was it the dried form, washed or in the solution form as synthesized. Without calcining, for sure the will be SI extract (capping agent) present during the catalysis. That implies that we cannot say for sure that the observed catalytic observation is solely due to the Cu2O or the capping agent. To dissociate the any influence of the capping agent, there should be a blank experiment, with a heated SI extract without Cu, just in case the as synthesized Cu2O was not calcined or at least the capping agent was not washed away and dried otherwise the effect of the extract on the reaction is not accounted for.

Results and discussion:

In the TEM, there are larger particles 50 nm and also several smaller range , 7 nm. On the larger particles, an SAED was performed showing semi crystalline materials, can the authors include the planes, which can be obtained from the lattice d-spacing? This will help to ascertain the state of the Cu at least during the TEM measurement. Depending on the prior treatment to the TEM measurement, it could help explain the size discrepancy comparing to the XRD. 

Secondly on the TEM, based on how many particles did the authors conclude that the range of the particle size is 5-7 nm. And did they include the larger particles?

On the XRD measurement, very sharp peak (rather than broad peaks for nanoparticles) and high degree of disorder and amorphous background is observed. This explains why removing the capping agent was necessary (see my comment at the experimental section above). Going forward, could the authors describe the drying process employed before the XRD was measured?

Also, i will like the authors to comment further, why the particles could grow to larger crystals, more than 6 times it size during the drying process, provided that, the Cu2O was synthesized at 90 oC for 5 h. 

Author Response

Reply to reviewer 2

Reviewer 3 Report

Manuscript title: Andean Sacha inchi (Plukenetia volubilis L.) leaf mediated synthesis of Cu2O nanoparticles: Low cost approach

Manuscript ID: bioengineering-814361

In this work, Brajesh Kumar et al used Andean Sacha inchi (Plukenetia volubilis L.) leaves as a bioreductant to synthesize Cu2O nanoparticles under heating. The authors have used different nanoparticle characterization techniques to support their synthetic method and hypothesis. They have used different spectroscopic methods such as UV-Vis, DLS, FTIR and fluorescence to understand about the synthesized particles and also used electron microscopy to investigate their morphology. XRD was performed to investigate into the crystallinity.

In addition, the authors have explored photocatalytic degradation activity of Cu2O nanoparticles against methylene blue. This is a good paper and can be of use by the readers of bioengineering. Hence I recommend publication of the article in bioengineering after revision.

1)       The authors should expand the introduction and include different bioreductants used for synthesis of nanoparticles. Also, the authors claimed Cu2O nanoparticle has not been synthesized using Andean Sacha inchi leaves. The authors should make a paragraph in the introduction section, compiling literature that reports nanoparticle synthesis using Andean Sacha inchi leaves and Andean Sacha inchi leaves used for other nanoparticle synthesis and their application. It will be good for the readers to know the background and important of the leaves for nanoparticle synthesis. In addition, the authors should also elaborate different routes for synthesis of Cu2O nanoparticles reported in literature and how the present synthetic route is different and better.

2)       The authors should note different font size used in same text. For e.g. in experimental section (2.1) why is ‘Copper Nitrate (Cu (NO3)2.3H2O, 99.0 %) and Methylene Blue (MB, 99.5%) 75 were purchased from Spectrum, USA’ in a bigger font than rest of the text. Same goes for section 2.2, 2.3, 2.4 and in fact through out text. There is different font size used. (Also in Figure caption)

3)       If possible, the authors should provide a high magnification TEM image of the nanoparticles.

4)       The resolution of FTIR graph is very low, the authors should improve the figure resolution.

5)       In the DLS analysis, the authors found the size of the nanoparticles as 8.2 ± 2.1 nm. However, in the conclusion, the authors reported the size as 5-7 nm. What is the reason ? In the TEM analysis, the authors found the particle size 5-7 nm. How ? Did the authors individually measured the diameter of nanoparticles or run a size analysis. If yes, please provide the data.

Round 2

Reviewer 1 Report

The authors have done commendable job in addressing the concerns of the reviewers. The manuscript should be accepted for publishing in bioengineering.